# ADVERSARIAL ATTACKS ON NODE EMBEDDINGS

## ABSTRACT

The goal of network representation learning is to learn low-dimensional node embeddings that capture the graph structure and are useful for solving downstream tasks. However, despite the proliferation of such methods, there is currently no study of their robustness to adversarial attacks. We provide the first adversarial vulnerability analysis on the widely used family of methods based on random walks. We derive efficient adversarial perturbations that poison the network structure and have a negative effect on both the quality of the embeddings and the downstream tasks. We further show that our attacks are transferable since they generalize to many models, and are successful even when the attacker is restricted.

## 1 INTRODUCTION

Unsupervised node embedding (network representation learning) approaches are becoming increasingly popular and achieve state-of-the-art performance on many network learning tasks (Cai et al., 2017). The goal is to embed each node in a low-dimensional feature space such that the graph's structure is captured. The learned embeddings are subsequently used for downstream tasks such as link prediction, node classification, community detection, and visualization. Among the variety of proposed approaches, techniques based on random walks (RWs) (Perozzi et al.; Grover & Leskovec) are highly successful since they incorporate higher-order relational information. Given the increasing popularity of these method, there is a strong need for an analysis of their robustness. In particular, we aim to study the existence and effects of *adversarial perturbations*. A large body of research shows that traditional (deep) learning methods can easily be fooled/attacked: even slight deliberate data perturbations can lead to wrong results (Goodfellow et al., 2014; Mei & Zhu, 2015; Carlini & Wagner, 2017; Liang et al., 2017; Cisse et al., 2017; Lin et al., 2017; Chen et al., 2017a).

So far, however, the question of adversarial perturbations for node embeddings has not been addressed. This is highly critical, since especially in domains where graph embeddings are used (e.g. the web) adversaries are common and false data is *easy to inject*: e.g. spammers might create fake followers on social media or fraudsters might manipulate friendship relations in social networks. Can node embedding approaches be easily fooled? The answer to this question is not immediately obvious. On one hand, the relational (non-i.i.d.) nature of the data might improve robustness since the embeddings are computed for all nodes jointly rather than for individual nodes in isolation. On the other hand, the propagation of information might also lead to cascading effects, where perturbations in one part of the graph might affect many other nodes in another part of the graph.

Compared to the existing works on adversarial attacks our work significantly differs in various aspects. First, by operating on plain graph data, we do not perturb the features of individual instances but rather their interaction/dependency structure. Manipulating the structure (the graph) is a highly realistic scenario. For example, one can easily add or remove fake friendship relations on a social network, or write fake reviews to influence graph-based recommendation engines. Second, the node embedding works are typically trained in an unsupervised and transductive fashion. This means that we cannot rely on a single end-task that our attack might exploit to find appropriate perturbations, and we have to handle a challenging poisoning attack where the model is learned *after* the attack. That is, the model cannot be assumed to be static as in most other adversarial attack works. Lastly, since graphs are discrete classical gradient-based approaches (Li et al.; Mei & Zhu, 2015) for finding adversarial perturbations that were designed for continuous data are not well suited. Particularly for RW-based methods, the gradient computation is not directly possible since they are based on a non-differentiable sampling procedure. How to design efficient algorithms that are able to find adversarial perturbations in such a challenging – discrete and combinatorial – graph domain?

We propose a principled strategy for adversarial attacks on unsupervised node embeddings. Exploiting results from eigenvalue perturbation theory (Stewart, 1990) we are able to efficiently solve a challenging bi-level optimization problem associated with the poisoning attack. We assume an attacker with full knowledge about the data and the model, thus, ensuring reliable vulnerability analysis in the worst case. Nonetheless, our experiments on transferability demonstrate that our strategy generalizes – attacks learned based on one model successfully fool other models as well.

Overall, we shed light on an important problem that has not been studied so far. We show that node embeddings are sensitive to adversarial attacks. Relatively few changes are needed to significantly damage the quality of the embeddings even in the scenario where the attacker is restricted. Furthermore, our work highlights that more work is needed to make node embeddings robust to adversarial perturbations and thus readily applicable in production systems.

## 2 RELATED WORK

We focus on adversarial attacks on unsupervised node embedding approaches based on random walks (RWs), and further show how one can easily apply a similar analysis to attack other node embeddings based on factorization. For a recent extensive survey, also of other non-RW based approaches, we refer to Cai et al. (2017). Moreover, while many (semi-)supervised learning methods (Kipf & Welling, 2016; Defferrard et al.) have been introduced, we focus on unsupervised methods since they are often used in practice due to their flexibility in solving various downstream tasks.

**Adversarial attacks.** Attacking machine learning models has a long history, with seminal works on SVMs and logistic regression (Biggio et al., 2012; Mei & Zhu, 2015). Deep neural networks were also shown to be highly sensitive to small adversarial perturbations to the input (Szegedy et al., 2013; Goodfellow et al., 2014). While most works focus on image classification, recent works have shown the existence of adversarial examples also in other domains (Liang et al., 2017; Grosse et al.).

Different taxonomies exist characterizing the attacks/adversaries based on their goals, knowledge, and capabilities (Biggio et al.; Papernot et al.; Muñoz-González et al., 2017). The two dominant attacks types are poisoning attacks that target the training data (the model is trained *after* the attack) and evasion attacks that target the test data/application phase (the learned model is assumed fixed). Compared to evasion attacks, poisoning attacks are far less studied (Koh & Liang, 2017; Muñoz-González et al., 2017; Li et al.; Mei & Zhu, 2015; Chen et al., 2017a) since they require solving a challenging bi-level optimization problem.

**Attacks on semi-supervised graph models.** The robustness of semi-supervised graph classification methods to adversarial attacks has recently been analyzed (Zügner et al., 2018; Dai et al., 2018a). The first work, introduced by Zügner et al. (2018), linearizes a graph convolutional network (GCN) (Kipf & Welling, 2016) to derive a closed-form expression for the change in class probabilities for a given edge/feature perturbation. They calculate a score for each possible edge flip based on the classification margin and greedily pick the top edge flips with highest scores. Later, Dai et al. (2018a) proposed a reinforcement (Q-)learning formulation where they decompose the selection of relevant edge flips into selecting the two end-points. Both approaches focus on targeted attacks (misclassify a given node) for the semi-supervised graph classification task. In contrast, our work focuses on general attacks (decrease the overall quality) on unsupervised node embeddings.

**Manipulating graphs.** In the context of graph clustering, Chen et al. (2017b) measure the changes in the result when injecting noise to a bi-partite graph of DNS queries, but do not focus on automatically generating attacks. There is an extensive literature on works that optimize the graph structure to manipulate e.g. information spread in a network (Chen et al.; Khalil et al.), user opinions (Amelkin & Singh, 2017; Chaoji et al.), shortest paths (Phillips; Israeli & Wood), page rank scores and other metrics (Avrachenkov & Litvak; Chan et al.). Remotely related are poisoning attacks on multi-task relationship learning (Zhao et al., 2018). While they exploit the relations between different tasks, they still deal with the classical scenario of i.i.d. instances within each task.

**Robustness and adversarial training.** The robustification of machine learning models has also been studied – known as adversarial machine learning or robust machine learning. Such approaches are out of scope for this paper and we do not discuss them. The goal of adversarial training (e.g. via GANs (Dai et al., 2018b)) is to improve the embeddings, while our goal is to damage the embeddings produced by existing models by perturbing the graph structure.

## 3 ATTACKING NODE EMBEDDINGS

Here we explore poisoning attacks on the graph structure – the attacker is capable of adding or removing (flipping) edges in the original graph within a given budget. We focus mainly on approaches based on random walks and extend the analysis to spectral approaches (Sec. 6.2 in the appendix).

### 3.1 BACKGROUND AND PRELIMINARIES

Let $G = (V, E)$ be an undirected unweighted graph where $V$ is the set of nodes, $E$ is the set of edges, and $A \in \{0, 1\}^{|V| \times |V|}$ is the adjacency matrix. The goal of network representation learning is to find a low-dimensional embedding $z_v \in \mathbb{R}^K$ for each node with $K \ll |V|$. This dense low-dimensional representation should preserve information about the network structure – nodes similar in the original network should be close in the embedding space. DeepWalk (Perozzi et al.) and node2vec (Grover & Leskovec) learn an embedding based on RWs by extending and adapting the skip-gram architecture (Mikolov et al., 2013) for learning word embeddings. They sample finite (biased) RWs and use the co-occurrence of node-context pairs in a given window in each RW as a measure of similarity. To learn $z_v$ they maximize the probability of observing $v$'s neighborhood.

### 3.2 ATTACK MODEL

We denote with $\hat{A}$ the adjacency matrix of the graph obtained after the attacker has modified certain entries in $A$. We assume the attacker has a given, fixed budget and is only capable of modifying $f$ entries, i.e. $||\hat{A} - A||_0 = 2f$ (we have $2f$ since $G$ is undirected). The goal of the attacker is to damage the quality of the learned embeddings, which in turn harms subsequent learning tasks such as node classification or link prediction that use the embeddings as features. We consider both a general attack that aims to degrade the embeddings of the network as a whole, as well as a targeted attack that aims to damage the embedding regarding a specific target or specific task.

The quality of the embeddings is measured by the loss $\mathcal{L}(A, Z)$ of the model under attack, with lower loss corresponding to higher quality, where $Z \in \mathbb{R}^{N \times K}$ is the matrix containing the embeddings of all nodes. Thus, the goal of the attacker is to *maximize* the loss. We can formalize this as the following bi-level optimization problem:

$$\hat{A}^* = \arg \max_{\hat{A} \in \{0,1\}^{N \times N}} \mathcal{L}(\hat{A}, Z^*) \quad Z^* = \min_Z \mathcal{L}(\hat{A}, Z) \quad \text{subj. to } ||\hat{A} - A||_0 = 2f \, , \hat{A} = \hat{A}^T \quad (1)$$

Here, $Z^*$ is always the *'optimal'* embedding resulting from the (to be optimized) graph $\hat{A}$, i.e. it minimizes the loss, while the attacker tries to maximize the loss. Solving such a problem is highly challenging given its discrete and combinatorial nature, thus we derive efficient approximations.

### 3.3 GENERAL ATTACK

Since the first step in the embedding approaches is to generate a set of random walks that serve as a training corpus for the skip-gram model, the bi-level optimization problem is even more complicated. We have $Z^* = \min_Z \mathcal{L}(\{r_1, r_2, \dots\}, Z)$ with $r_i \sim RW_l(\hat{A})$, where $RW_l$ is an intermediate stochastic procedure that generates RWs of length $l$ given the graph $\hat{A}$ which we are optimizing. By flipping (even a few) edges in the original graph, the attacker necessarily changes the set of possible RWs, thus changing the training corpus. Therefore, this RW generation process precludes any gradient-based methods. To tackle this challenge we leverage recent results that show that (given certain assumptions) RW based node embedding approaches are implicitly factorizing the Pointwise Mutual Information (PMI) matrix (Yang & Liu, 2015; Qiu et al., 2017). We study DeepWalk as an RW-based representative approach since it's one of the most popular methods and has many extensions. Specifically, we use the results from Qiu et al. (2017) to sidestep the RW stochasticity.

**Lemma 1** (Qiu et al. (2017)). *DeepWalk is equivalent to factorizing $\tilde{M} = \log(\max(M, 1))$ with*

$$M = \frac{vol(A)}{T \cdot b} S, \quad \text{where} \quad S = \left( \sum_{r=1}^{T} P^r \right) D^{-1}, \quad \text{where} \quad P = D^{-1} A \quad (2)$$

*where the embedding $Z^*$ is obtained by the Singular Value Decomposition of $\tilde{M} = U \Sigma V^T$ using the top-$K$ largest singular values / vectors, i.e. $Z^* = U_K \Sigma_K^{1/2}$.*

Here, $D$ is the diagonal degree matrix with $D_{ii} = \sum_j A_{ij}$, $T$ is the window size, $b$ is the number of negative samples and $vol(A) = \sum_{i,j} A_{ij}$ is the volume. Since $M$ is sparse and has many zero entries the matrix $\log(M)$ where the $\log$ is elementwise is ill-defined and dense. To cope with this, similar to the Shifted Positive PMI (PPMI) approach the elementwise maximum is introduced to form $\tilde{M}$. Using this insight, we see that DeepWalk is equivalent to optimizing $\min_{\tilde{M}_K} ||\tilde{M} - \tilde{M}_K||_F^2$ where $\tilde{M}_K$ is the best rank-$K$ approximation to $\tilde{M}$. This in turn means that the loss for DeepWalk when using the *optimal* embedding $Z^*$ for a given graph $A$ is $\mathcal{L}_{DW_1}(A, Z^*) = \sqrt{\sum_{p=K+1}^{|V|} \sigma_p^2}$ where $\sigma_p$ are the singular values of $\tilde{M}(A)$ ordered decreasingly $\sigma_1 \geq \sigma_2 \cdots \geq \sigma_{|V|}$. This result shows that we do not need to construct random walks, nor do we have to (explicitly) learn the embedding $Z^*$ – it is implicitly considered via the singular values of $\tilde{M}(A)$. Accordingly, we have transformed the bi-level problem into a single-level optimization problem. However, maximizing $\mathcal{L}_{DW_1}$ is still challenging due to the singular value decomposition and the discrete nature of the problem.

**Gradient based approach.** Maximizing $\mathcal{L}_{DW_1}$ with a gradient-based approach is not straightforward since we cannot easily backpropagate through the SVD. To tackle this challenge we exploit ideas from eigenvalue perturbation theory (Stewart, 1990) to approximate $\mathcal{L}_{DW_1}(A)$ in closed-form without needing to recompute the SVD. This enables us to efficiently calculate the gradient.

**Theorem 1.** *Let $A$ be the initial adjacency matrix and $\tilde{M}(A)$ be the respective co-occurrence matrix. Let $u_p$ be the $p$-th eigenvector corresponding to the $p$-th largest eigenvalue of $\tilde{M}$. Given a perturbed matrix $A'$, with $A' = A + \Delta A$, and the respective change $\Delta \tilde{M}$. We can approximately compute the loss: $\mathcal{L}_{DW_1}(A') \approx \sqrt{\sum_{p=K+1}^{N} \left( u_p^T(\tilde{M} + \Delta \tilde{M})u_p \right)^2} =: \mathcal{L}_{DW_2}(A')$ and the approximation error is bounded by $|\mathcal{L}_{DW_1}(A') - \mathcal{L}_{DW_2}(A')| \leq ||\Delta \tilde{M}||_F$.*

The proof is given in the appendix. For a small $\Delta A$ and thus small $\Delta \tilde{M}$ we obtain a very good approximation, and if $\Delta A = \Delta \tilde{M} = 0$ then the loss is exact. Intuitively, we can think of using eigenvalue perturbation as analogous to taking the gradient of the loss w.r.t. $\tilde{M}(A)$. Now, gradient-based optimization is efficient since $\nabla_A \mathcal{L}_{DW_2}(A)$ avoids recomputing the eigenvalue decomposition. The gradient provides useful information for a small $\epsilon$ change, however, here we are considering discrete flips, i.e. $\epsilon = \pm 1$ so its usefulness is limited. Furthermore, using gradient-based optimization requires a dense instantiation of the adjacency matrix, which has complexity $O(N^2)$ in both runtime and memory (infeasible for large graphs). This motivates the need for our more advanced approach.

**Sparse closed-form approach.** Our goal is to efficiently compute the change in the loss $\mathcal{L}_{DW_1}(A)$ given a set of flipped edges. To do so we will analyze the change in the spectrum of some of the intermediate matrices and then derivate a bound on the change in the spectrum of the co-occurrence matrix, which in turn will give an estimate of the loss. First, we need some results.

**Lemma 2.** *The matrix $S$ in Eq. 2 is equal to $S = U(\sum_{r=1}^T \Lambda^r)U^T$ where the matrices $U$ and $\Lambda$ contain the eigenvectors and eigenvalues solving the generalized eigen-problem $Au = \lambda Du$.*

The proof is given in the appendix. We see that the spectrum of $S$ (and, thus, the one of $M$ by taking scalars into account) is obtainable from the generalized spectrum of $A$. The difference to Qiu et al. (2017)'s derivation where a factorization of $S$ using $A_{norm} := D^{-1/2}AD^{-1/2}$ is important. As we will show, our formulation using the generalized spectrum of $A$ is key for an efficient approximation.

Let $A' = A + \Delta A$ be the adjacency matrix after the attacker performed some edge flips. As above, by computing the generalized spectrum of $A'$, we can estimate the spectrum of the resulting $S'$ and $M'$. However, recomputing the eigenvalues $\lambda'$ of $A'$ for every possible set of edge flips is still not efficient for large graphs, preventing an effective application. Thus, we derive our first main result: an efficient approximation bounding the change in the singular values of $M'$ for any edge flip.

**Theorem 2.** *Let $\Delta A$ be a matrix with only 2 non-zero elements, namely $\Delta A_{ij} = \Delta A_{ji} = 1 - 2A_{ij}$ corresponding to a single edge flip $(i, j)$, and $\Delta D$ the respective change in the degree matrix, i.e. $A' = A + \Delta A$ and $D' = D + \Delta D$. Let $u_y$ be the $y$-th generalized eigenvector of $A$ with generalized eigenvalue $\lambda_y$. Then the generalized eigenvalue $\lambda'_y$ of $A'$ solving $\lambda'_y A' = \lambda'_y D' u'_y$ is approximately:*

$$\lambda'_y \approx \tilde{\lambda}'_y = \lambda_y + \Delta\lambda_y \qquad \Delta\lambda_y = \Delta w_{ij}(2u_{yi} \cdot u_{yj} - \lambda_y(u_{yi}^2 + u_{yj}^2)) \qquad (3)$$

*where $u_{yi}$ is the $i$-th entry of the vector $u_y$, and $\Delta w_{ij} = (1 - 2A_{ij})$ indicates the edge flip, i.e $\pm 1$.*

The proof is provided in the appendix. By working with the generalized eigenvalue problem in Theorem 2 we were able to express $A'$ and $D'$ after flipping an edge as *additive* changes to $A$ and $D$, this in turn enabled us to leverage results from eigenvalue perturbation theory to efficiently approximate the change in the spectrum. If we used $A_{norm}$ instead, the change to $A'_{norm}$ would be multiplicative preventing efficient approximations. Using Eq. 3, instead of recomputing $\lambda'$ we only need to compute $\Delta\lambda$, significantly reducing the complexity when evaluating different edge flips $(i,j)$. Using this result, we can now efficiently bound the change in the singular values of $S'$.

**Lemma 3.** *Let $A'$ be defined as before and $S'$ be the resulting matrix. The singular values of $S'$ are bounded: $\sigma_p(S') \leq \tilde{\sigma}_p(i,j) := \frac{1}{d'_{min}} \cdot \left| \sum_{r=1}^{T}(\tilde{\lambda'_{\pi(p)}})^r \right|$ where $\pi$ is a permutation simply ensuring that the final $\tilde{\sigma}_p(i,j)$ are sorted decreasingly, where $d'_{min}$ is the smallest degree in $A'$.*

We provide the proof in the appendix. Using this result, we can efficiently compute the loss for a rank-$K$ approximation/factorization of $M'$, which we would obtain when performing the edge flip $(i,j)$, i.e. $\mathcal{L}_{DW_3}(A') = \frac{vol(A)+2\Delta w_{ij}}{T \cdot b} \left[ \sum_{p=K+1}^{|V|} \tilde{\sigma}_p(i,j)^2 \right]^{1/2}$. While the original loss $\mathcal{L}_{DW_1}$ is based on the matrix $\tilde{M} = \log(\max(M,1))$, there are unfortunately currently no tools available to analyze the spectrum of $\tilde{M}$ given the spectrum of $M$. Therefore, we use $\mathcal{L}_{DW_3}$ as a surrogate loss for $\mathcal{L}_{DW_1}$ (Yang et al. similarly exclude the element-wise logarithm). As our experimental analysis shows, the surrogate loss is effective and we are able to successfully attack the node embeddings that factorize the actual co-occurrence matrix $\tilde{M}$, as well as the original skip-gram model. Similarly, methods based on spectral embedding, factorize the graph Laplacian and have a strong connection to the RW based approaches. We provide a similar detailed analysis in the appendix (Sec. 6.2).

**The overall algorithm.** Our goal is to maximize $\mathcal{L}_{DW_3}$ by performing $f$ edge flips. While Eq. 3 enables us to efficiently compute the loss for a single edge, there are still $\mathcal{O}(n^2)$ possible flips. To reduce the complexity when adding edges (see Sec. 4.2 for removing) we instead form a candidate set by randomly sampling $C$ candidate flips. This introduces a further approximation that nonetheless works well in practice. For every candidate we compute its impact on the loss via $\mathcal{L}_{DW_3}$ and greedily choose the top $f$ flips.[1] The runtime complexity of our overall approach is: $\mathcal{O}(N \cdot |E| + C \cdot N \log N)$. First, we can compute the generalized eigenvectors of $A$ in a sparse fashion in $\mathcal{O}(N \cdot |E|)$. Then we sample $C$ candidate edges, and for each we can compute the approximate eigenvalues in constant time (Theorem 2). To obtain the final loss, we sort the values leading to the overall complexity. The approach is easily parallelizable since every candidate edge flip can be evaluated in parallel.

### 3.4 TARGETED ATTACK

If the goal of the attacker is to attack a specific node $t \in V$, called the target, or a specific downstream task, it is suboptimal to maximize the overall loss via $L_{DW_*}$. Rather, we should define some other *target specific* loss that depends on $t$'s embedding – replacing the loss function of the *outer* optimization in Eq. 1 by another one operating on $t$'s embedding. Thus, for any edge flip $(i,j)$ we now need the change in $t$'s embedding – meaning changes in the eigen*vectors* – which is inherently more difficult to compute compared to changes in eigen/singular-*values*. We study two cases: misclassifying a target node and manipulating the similarity of node pairs (i.e. link prediction task).

**Surrogate embeddings.** To efficiently compute the change in eigenvectors, we define surrogate embeddings $\bar{Z}^*$. Specifically, instead of performing an SVD decomposition on $M$ (or equivalently $S$ with upscaling) and using the results from Lemma 2 we define $\bar{Z}^* = U(\sum_{r=1}^{T} \Lambda^r)$. Experimentally, using $\bar{Z}^*$ instead of $Z^*$ as the embedding showed no significant change in the performance on downstream tasks (even on the clean graph; suggesting its general use since it is more efficient to compute). Now, we can approximate the generalized eigenvectors, and thus $\bar{Z}^*(A')$, in closed-form:

**Theorem 3.** *Let $\Delta A, \Delta D$ and $\Delta w_{ij}$ be defined as before, and $\Delta\lambda_y$ be the change in the y-th generalized eigenvalue $\lambda_y$ as derived in Theorem 2. Then, the y-th generalized eigenvector $u'_y$ of $A'$ after performing the edge flip $(i,j)$ can be approximated with:*

$$u'_y \approx u_y - \Delta w_{ij}(A - \lambda D)^+(-\Delta\lambda_y u_y \circ d + E_i(u_{yj} - \lambda_y u_{yi}) + E_j(u_{yi} - \lambda_y u_{yj})) \quad (4)$$

*where $E_i(x)$ returns a vector of zeros except at position $i$ where the value is $x$, $d$ is a vector of the node degrees, $\circ$ is the Hadamard product, and $(\cdot)^+$ is the pseudo inverse.*

---

[1] A greedy approach where we sequentially flip the single best candidate, followed by a periodic recomputation of the eigenvalues did not show benefits, nor did an evolutionary strategy using Theorem 2 as a heuristic.

We provide the proof in the appendix. Computing Eq. 4 seems expensive at first due to the pseudo inverse term. However, note that this term does not depend on the particular edge flip we perform. Thus, we can pre-compute it once and furthermore, parallelize the computation for each $y$. Similarly, we can pre-compute $u_y d$, while the rest of the terms are all computable in $O(1)$. For any edge flip we can now efficiently compute the optimal embedding $\bar{Z}^*(A')$ using Eqs. 3 and 4. The t-th row of $\bar{Z}^*(A')$ is the desired embedding for a target node $t$ after the attack.

**Targeting node classification.** The goal is to enforce misclassification of the target $t$ for the downstream task of node classification (i.e. node labels are partially given). To fully specify the targeted attack we need to define the candidate flips and the target-specific loss responsible for scoring the candidates. As candidates we use $\{(v, t)|v \neq t\}$. For the loss, we first pre-train a classifier $\mathcal{C}$ on the clean embedding $\bar{Z}^*$. Then we predict the class probabilities $p_t$ of the target $t$ using the compromised $\bar{Z}^*_{t,\cdot}$ and we calculate the classification margin $m(t) = p_{t,c(t)} - \max_{c \neq c(t)} p_{t,c}$, where $c(t)$ is the ground-truth class for $t$. That is, our loss is the difference between the probability of the ground truth and the next most probable class after the attack. Finally, we select the top $f$ flips with smallest margin $m$ (note when $m(t) < 0$ node $t$ is misclassified). In practice, we average over 10 randomly trained classifiers. Another (future work) approach is to treat this as a tri-level optimization problem.

**Targeting link prediction.** The goal of the attack is: given a set of target node pairs $\mathcal{T} \subset V \times V$, decrease the similarity between the nodes that have an edge, and increase the similarity between nodes that do not have an edge, by modifying *other* parts of the graph – i.e. it is not allowed to directly flip pairs in $\mathcal{T}$. For example, in an e-commerce graph representing users and items, the goal might be to increase the similarity between a certain item and user, by adding/removing connections between other users/items. To achieve this, we first train the initial clean embedding without the target edges. Then, for a candidate set of flips, we estimate $\bar{Z}^*$ using Eqs. 3 and 4 and use them to calculate the average precision score (AP score) on the target set $\mathcal{T}$, with $\bar{Z}^*_i (\bar{Z}^*_j)^T$ as a similarity measure. Finally, we pick the top $f$ flips with lowest AP scores and use them to poison the network.

## 4 EXPERIMENTAL EVALUATION

Since this is the first work considering adversarial attacks on node embeddings there are no known baselines. Similar to works that optimize the graph structure (Chen et al.) we compare with several strong baselines. $\mathcal{B}_{rnd}$ randomly flips edges (we report averages over ten seeds), $\mathcal{B}_{eig}$ removes edges based on their eigencentrality in the line graph $L(A)$, and $\mathcal{B}_{deg}$ removes edges based on their degree centrality in $L(A)$ – or equivalently sum of degrees in the original graph. When adding edges we use the same baselines as above, now calculated on the complement graph, except for $\mathcal{B}_{eig}$ since it is infeasible to compute even for medium size graphs. $\mathcal{A}_{DW_2}$ denotes our gradient based attack, $\mathcal{A}_{DW_3}$ our closed-form attack, $\mathcal{A}_{link}$ our link prediction attack, $\mathcal{A}_{class}$ our node classification attack. The size of the sampled candidate set for adding edges is 20K (for removing edges see Sec. 4.2).

We aim to answer the following questions: (Q1) how good are our approximations of the loss; (Q2) how much damage is caused to the embedding quality by our attacks/baselines; (Q3) can we still perform a successful attack when restricted; (Q4) what characterizes selected (top) adversarial edges; (Q5) how do the targeted attacks affect downstream tasks; and (Q6) are the attacks transferable.

We set DeepWalk's hyperparameters to: $T = 5, b = 5, K = 64$ and use a logistic regression for classification. We analyze three datasets: Cora ($N = 2810, |E| = 15962$, McCallum et al.), Citeseer ($N = 2110, |E| = 7336$, Giles et al.), and PolBlogs ($N = 1222, |E| = 33428$, Adamic & Glance (2005)). In all experiments, after choosing the top $f$ flips we retrain the embeddings and report the final performance since this is a poisoning attack. Note, for the *general attack*, the downstream node classification task is *only a proxy* for estimating the quality of the embeddings after the attack, it is not our goal to damage this task, but rather to attack the unsupervised embeddings in general.

### 4.1 APPROXIMATION QUALITY

To estimate the approximation quality we randomly select a subset of 20K candidate flips and compute the correlation between the actual loss and our approximation as measured by Pearson's $R$ score. For example, for $K = 32$ we have $R(\mathcal{L}_{DW_2}, \mathcal{L}_{DW_1}) = 0.11$ and $R(\mathcal{L}_{DW_3}, \mathcal{L}_{DW_1}) = 0.90$, clearly showing that our closed-form strategy approximates the loss significantly better compared to the gradient-based one. Similarly, $\mathcal{L}_{DW_3}$ is a better approximation than $\mathcal{L}_{DW_2}$ for $K = 16, 64, 128$.

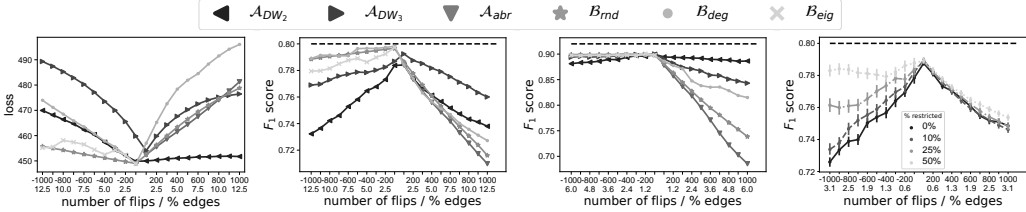

Figure 1: Vulnerability of the embeddings under the general attack for increasing number of flips. The dotted line shows the performance before attacking.

## 4.2 GENERAL ATTACK

To obtain a better understanding we investigate the effect of removing and adding edges separately. Since real graphs are usually sparse, for removing we set the candidate set to be the set of all edges, with one edge set aside for each node to ensure we do not have singleton nodes. To obtain candidate edges for adding we randomly sample a set of edges. We then simply select the top $f$ edges from the candidate set according to our scoring function. For adding edges, we also implemented an alternative add-by-remove strategy denoted as $\mathcal{A}_{abr}$. Here, we first add $cf$-many edges randomly sampled from the candidate set to the graph and subsequently remove $(c-1)f$-many of them. This strategy performed better empirically. Since the graph is undirected, for each $(i, j)$ we also flip $(j, i)$.

Fig. 1 answers question (Q2). Removed/added edges are denoted on the x-axis with negative/positive values respectively. On Fig. 1a we see that our strategies achieve a significantly higher loss compared to the baselines when removing edges. To analyze the change in the embedding quality we consider the node classification task (i.e. using it as a proxy to evaluate quality; this is *not* our targeted attack). Interestingly, $\mathcal{B}_{deg}$ is the strongest baseline w.r.t. to the loss, but this is not true for the downstream task. As shown in Fig. 1b and 1c, our strategies significantly outperform the baselines. As expected, $\mathcal{A}_{DW_3}$ and $\mathcal{A}_{abr}$ perform better than $\mathcal{A}_{DW_2}$. On Cora our attack can cause up to around 5% *more* damage compared to the strongest baseline. On PolBlogs, by adding only 6% edges we can decrease the classification performance by more than 23%, while being more robust to removing edges.

**Restricted attacks.** In the real world, attackers cannot attack any node, but rather only specific nodes under their control, which translates to restricting the candidate set. To evaluate the restricted scenario, we first initialize the candidate sets as before, then we randomly choose a given percentage $p_r$ of nodes as restricted and discard every candidate that includes them. As expected, the results in Fig. 1d show that for increasingly restrictive sets with $p_r = 10\%, 25\%, 50\%$, our attack is able to do less damage. However, we always outperform the baselines (not plotted), and even in the case when half of the nodes are restricted ($p_r = 50\%$) we are still able to damage the embeddings. With this we are can answer question (Q3) affirmatively – the attacks are successful even when restricted.

**Analysis of selected adversarial edges.** In Fig. 2a we analyze the top 1K edges on Cora-ML. For each edge we consider its source node degree (destination node, resp.) and plot it on the x-axis (y-axis). The heatmap shows adversarial edge counts divided by total edge counts for each bin. We see that low, medium and high degree nodes are all represented. In Fig. 2b we plot the edge centrality distribution for the top 1K adversarial edges and compare it with the distribution of the remaining edges. There is no clear distinction. The findings highlight the need for a principled method such as ours since using intuitive heuristics such as degree/edge centrality cannot identify adversarial edges.

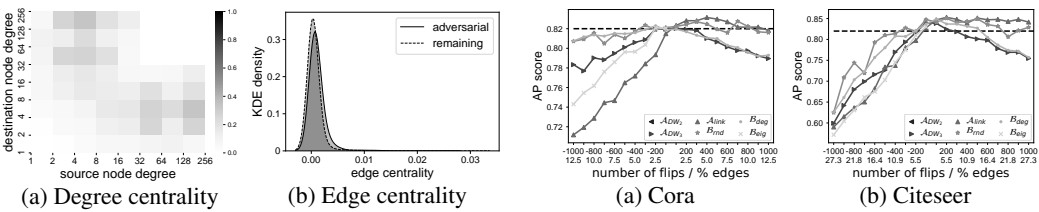

Figure 2: Analysis of the adversarial edges.    Figure 3: Targeted attack on the link prediction

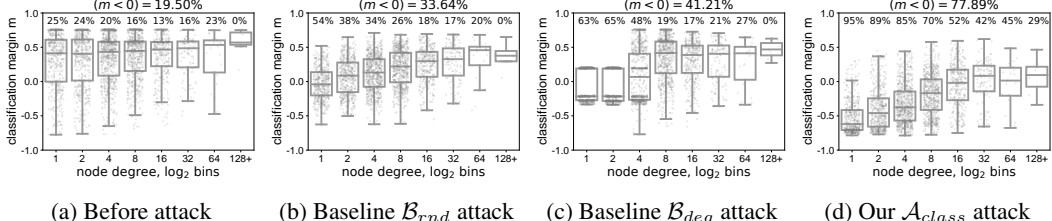

|           |           |           |           |
|-----------|-----------|-----------|-----------|
| (a) Before attack | (b) Baseline $\mathcal{B}_{rnd}$ attack | (c) Baseline $\mathcal{B}_{deg}$ attack | (d) Our $\mathcal{A}_{class}$ attack |

Figure 4: Margin distribution for different attacks binned according to their degrees (lower is better).

## 4.3 TARGETED ATTACK

To obtain a better understanding of the performance we study the margin $m(t)$ before and after the attack considering every node $t$ as a potential target. We allow only $(d_t + 3)$ flips for attacking each node ensuring the degrees stay similar. Each dot in Fig. 4 represents one node grouped by its degree in the clean graph (logarithmic bins). We see that low-degree nodes are easier to misclassify ($m(t) < 0$), and that high degree nodes are more robust in general – the baselines have $0\%$ success. Our method, however, can successfully attack even high degree nodes. In general, our attack is significantly more effective across all bins – as shown by the numbers on top of each box – with $77.89\%$ nodes successfully misclassified on average compared to e.g. only $33.64\%$ for $\mathcal{B}_{rnd}$ . For the link prediction task (Fig. 3) we are similarly able to cause significant damage – e.g. $\mathcal{A}_{link}$ achieves almost $10\%$ decrease in performance by flipping around $12.5\%$ of edges on Cora, significantly better than all other baselines. Here again, compared to adding edges, removing has a stronger effect. Overall, answering (Q5), both experiments confirm that our attacks hinder the downstream tasks.

## 4.4 TRANSFERABILITY

The question of transferability – do attacks learned for one model generalize to other models – is important since in practice the attacker might not know the model used by the system under attack. However, if transferability holds, such knowledge is not required. To obtain the perturbed graph, we remove the top $f$ adversarial edges with the $\mathcal{A}_{DW_3}$ attack. The same perturbed graph is then used to learn node embeddings using several other state-of-the-art approaches. Table 1 shows the change in node classification performance compared to the embeddings learned on the clean graph for each method respectively. We tune the key hyperparameters for each method (e.g. $p$ and $q$ for node2vec).

Table 1: Transferability: The change in $F_1$ score (in percent) compared to the clean/original graph.

| Cora / Citeseer | DeepWalk (SVD) | DeepWalk (SGNS) | node2vec | Spect. Embd. | Label Prop. | GCN |
|-----------------|----------------|-----------------|----------|--------------|-------------|-----|
| $f = 250(03.1\%)$ | -3.59 | -3.97 | -2.04 | -2.11 | -5.78 | -3.34 |
| $f = 500(06.3\%)$ | -5.22 | -4.71 | -3.48 | -4.57 | -8.95 | -2.33 |
| $f = 250(06.8\%)$ | -7.59 | -5.73 | -6.45 | -3.58 | -4.99 | -2.21 |
| $f = 500(13.6\%)$ | -9.68 | -11.47 | -10.24 | -4.57 | -6.27 | -8.61 |

Answering (Q6), the results show that our attack generalizes: the adversarial edges have a noticeable impact on other models as well. We can damage DeepWalk trained with the skip-gram objective with negative sampling (SGNS) showing that the factorization analysis is successful. We can even damage the performance of semi-supervised approaches such as GCN and Label Propagation. Compared to the transferability of the baselines (Sec. 6.3) our attack causes significantly more damage.

## 5 CONCLUSION

We demonstrate that node embeddings are vulnerable to adversarial attacks which can be efficiently computed and have a significant negative effect on node classification and link prediction. Furthermore, successfully poisoning the system is possible with relatively small perturbations and under restriction. More importantly, our attacks generalize - the adversarial edges are transferable across different models. Future work includes modeling the knowledge of the attacker, attacking other network representation learning methods, and developing effective defenses against such attacks.

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

# 6 APPENDIX

## 6.1 PROOFS AND DERIVATIONS

*Proof.* **Theorem 1.** Applying eigenvalue perturbation theory we obtain that $\lambda'_p = \lambda_p + u_p^T(\Delta\tilde{M})u_p$ where $\lambda'_p$ is the eigenvalue of $\tilde{M}'$ obtained after perturbing a single edge based on $A'$. Using the fact that $\lambda_p = u_p^T\tilde{M}u_p$, and the fact that singular values are equal to the absolute value of the corresponding eigenvalues we obtain the desired result. $\qquad\square$

*Proof.* **Theorem 2.** Denote with $e_i$ the vector of all zeros and a single one at position $i$. Then, we have $\Delta A = \Delta w_{ij}(e_ie_j^T + e_je_i^T)$ and $\Delta D = \Delta w_{ij}(e_ie_i^T + e_je_j^T)$. From eigenvalue perturbation theory Stewart (1990), we get: $\lambda'_y \approx \lambda_y + u_y^T(\Delta A - \lambda_y\Delta D)u_y$. Substituting $\Delta A/\Delta D$ concludes the proof. $\qquad\square$

We include an intermediate result which is required for proving Lemma 2 and Lemma 3.

**Lemma 4.** $\lambda$ *is an eigenvalue of* $D^{-1/2}AD^{-1/2} := A_{norm}$ *with eigenvector* $\hat{u} = D^{1/2}u$ *if and only if* $\lambda$ *and* $u$ *solve the generalized eigen-problem* $Au = \lambda Du$.

*Proof.* **Lemma 4.** We have $Az = \lambda Dz \implies (Q^{-1}AQ^{-T})(Q^Tz) = \lambda(Q^Tz)$ for any real symmetric $A$ and any positive definite $D$, where $D = QQ^T$ using the Cholesky factorization. Substituting the adjacency/degree matrix and noticing that $Q = Q^T = D^{1/2}$ we obtain the result. $\qquad\square$

*Proof.* **Lemma 2.** $S$ is equal to a product of three matrices $S = D^{-1/2}\big(\hat{U}\big(\sum_{r=1}^T \hat{\Lambda}^r\big)\hat{U}^T\big)D^{-1/2}$ where $\hat{U}\hat{\Lambda}\hat{U}^T = D^{-1/2}AD^{-1/2} =: A_{norm}$ is the eigenvalue decomposition of $A_{norm}$ (Qiu et al. (2017)). From Lemma 4 we have the fact that $\lambda$ is an eigenvalue of $D^{-1/2}AD^{-1/2}$ with eigenvector $\hat{u} = D^{1/2}u$ if and only if $\lambda$ and $u$ solve the generalized eigen-problem $Au = \lambda Du$. Substituting $\hat{\Lambda} = \Lambda$ and $\hat{U} = D^{1/2}U$ in $S$, and since $D$ is diagonal, we obtain the result. $\qquad\square$

*Proof.* **Lemma 3.** Following Qiu et al. (2017), the singular values of $S$ can be bounded by $\sigma_p(S) \leq \frac{1}{d_{min}}\big|\sum_{r=1}^T (\hat{\mu}_{\pi(p)})^r\big|$ where $\mu$ are the (standard) eigenvalues of $A_{norm}$. Using Lemma 4, the same bound applies using the generalized eigenvalues $\lambda_p$ of $A$. Now using Theorem 2, we obtain $\tilde{\lambda}'_p$ an approximation of the p-th *generalized* eigenvalue of $A'$. Plugging it into the singular value bound we obtain: $\sigma_p(S) \leq \frac{1}{d_{min}}\big|\sum_{r=1}^T (\tilde{\lambda}'_{\pi(p)})^r\big|$ which concludes the proof. $\qquad\square$

Please note that the permutation $\pi$ does not need be computed/determined explicitly. In practice, for every $\tilde{\lambda}'_p$, we compute the term $\big|\sum_{r=1}^T (\tilde{\lambda}'_p)^r\big|$. Afterwards, these terms are simply sorted.

## 6.2 ANALYSIS OF SPECTRAL EMBEDDING METHODS

**Attacking spectral embedding.** Finding the spectral embedding is equivalent to the following trace minimization problem:

$$\min_{Z \in \mathbb{R}^{|V| \times K}} Tr(Z^T L_{xy} Z) = \sum_{i=1}^{K} \lambda_i(L_{xy}) = \mathcal{L}_{SC} \tag{5}$$

subject to orthogonality constraints, where $L_{xy}$ is the graph Laplacian. The solution is obtained via the eigen-decomposition of $L$, with $Z^* = U_K$ where $U_K$ are the $K$-first eigen-vectors corresponding to the $K$-smallest eigenvalues $\lambda_i$. The Laplacian is typically defined in three different ways: the unnormalized Laplacian $L = D - A$, the normalized random walk Laplacian $L_{rw} = D^{-1}L = I - D^{-1}A$ and the normalized symmetric Laplacian $L_{sym} = D^{-1/2}LD^{-1/2} = I - D^{-1/2}AD^{-1/2} = I - A_{norm}$, where $A, D, A_{norm}$ are defined as before.

**Lemma 5** (Von Luxburg). *$\lambda$ is an eigenvalue of $L_{rw}$ with eigenvector $u$ if and only if $\lambda$ is an eigenvalue of $L_{sym}$ with eigenvector $w = D^{1/2}u$. Furthermore, $\lambda$ is an eigenvalue of $L_{rw}$ with eigenvector $u$ if and only if $\lambda$ and $u$ solve the generalized eigen-problem $Lu = \lambda Du$.*

From Lemma 5 we see that we can attack both normalized versions of the graph Laplacian with a single attack strategy since they have the same eigenvalues. It also helps us to do that efficiently similar to our previous analysis (Theorem. 3).

**Theorem 4.** *Let $L_{rw}$ (or equivalently $L_{sym}$) be the initial graph Laplacian before performing a flip and $\lambda_y$ and $u_y$ be any eigenvalue and eigenvector of $L_{rw}$. The eigenvalue $\lambda'_y$ of $L'_{rw}$ obtained after flipping a single edge $(i, j)$ is*

$$\lambda'_y \approx \lambda_y + \Delta w_{ij}((u_{yi} - u_{yj})^2 - \lambda_y(u_{yi}^2 + u_{yj}^2)) \tag{6}$$

*where $u_{yi}$ is the $i$-th entry of the vector $u_y$.*

*Proof.* From Lemma 5 we can estimate the change in $L_{rw}$ (or equivalently $L_{sym}$) by estimating the eigenvalues solving the generalized eigen-problem $Lu = \lambda Du$. Let $\Delta L = L' - L$ be the change in the unnormalized graph Laplacian after performing a single edge flip $(i, j)$ and $\Delta D$ be the corresponding change in the degree matrix. Let $e_i$ be defined as before. Then $\Delta L = (1 - 2A_{ij})(e_i - e_j)(e_i - e_j)^T$ and $\Delta D = (1 - 2A_{ij})(e_i e_i^T + e_j e_j^T)$. Based on the theory of eigenvalue perturbation we have $\lambda'_y \approx \lambda_y + u_y^T(\Delta L - \lambda_y \Delta D)u_y$. Substituting $\Delta L$ and $\Delta D$ are re-arranging we get the above results. $\square$

Using now Theorem 4 and Eq. 5 we finally estimate the loss of the spectral embedding after flipping an edge $\mathcal{L}_{SC}(L'_{rw}, Z) \approx \sum_{p=1}^{K} \lambda'_p$. Note that here we are summing over the $K$-first *smallest* eigenvalues. We see that spectral embedding and the random walk based approaches are indeed very similar.

We provide similar analysis for the the unnormalized Laplacian:

**Theorem 5.** *Let $L$ be the initial unnormalized graph Laplacian before performing a flip and $\lambda_y$ and $u_y$ be any eigenvalue and eigenvector of $L$. The eigenvalue $\lambda'_y$ of $L'$ obtained after flipping a single edge $(i, j)$ can be approximated by:*

$$\lambda'_y \approx \lambda_y - (1 - 2A_{ij})(u_{yi} - u_{yj})^2 \tag{7}$$

*Proof.* Let $\Delta A = A' - A$ be the change in the adjacency matrix after performing a single edge flip $(i, j)$ and $\Delta D$ be the corresponding change in the degree matrix. Let $e_i$ be defined as before. Then $\Delta L = L' - L = (D + \Delta D) - (A + \Delta A) - (D - A) = \Delta D - \Delta A = (1 - 2A_{ij})(e_i e_i^T + e_j e_j^T - (e_i e_j^T + e_j e_i^T))$. Based on the theory of eigenvalue perturbation we have $\lambda'_y \approx \lambda_y + u_y^T(\Delta L)u_y$. Substituting $\Delta L$ and re-arranging we get the above results. $\square$

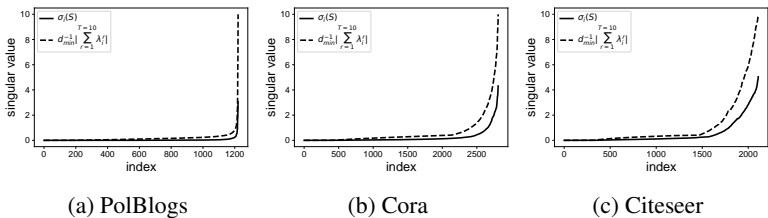

|           |           |           |
|:---------:|:---------:|:---------:|
| (a) PolBlogs | (b) Cora | (c) Citeseer |

Figure 5: The singular value of $S$ and our derived upper bound.

## 6.3 FURTHER EXPERIMENTAL EVIDENCE

**Upper bound on singular values.** From Lemma 3 we have that $\mathcal{L}_{DW3}$ is an upper bound on $\mathcal{L}_{DW1}$ (excluding the elementwise logarithm) so maximizing $\mathcal{L}_{DW3}$ is principled. To gain a better understanding of the tightness of the bound we visualize the singular values of $S$ and their respective upper-bound for all datasets. As we can see in Fig. 5, the gap is different for different datasets and relatively small. Furthermore we can notice that the gap tends to increase for larger singular values.

**Transferability of the baselines.** To further support the transferability of our proposed attack we also examine the transferability of the baseline attacks. Specifically, we examine the transferability of $\mathcal{B}_{eig}$ since it is the strongest baseline when removing edges as shown in Fig. 1b. We use the same experimental setup as in Sec. 4.4 and show the results in Table 2. We can see that compared to our proposed attack the baseline can do a significantly smaller amount of damage (compare to results in Table 1). Interestingly, it can do significant damage to GCN when removing 250 edges on Cora, but not when removing 500 edges. We plan on exploring this counterintuitive finding in future work.

Table 2: Transferability: The change in $F_1$ score (in percent) compared to the clean/original graph.

| Cora / Citeseer | DeepWalk (SVD) | DeepWalk (SGNS) | node2vec | Spect. Embd. | Label Prop. | GCN |
|---|---|---|---|---|---|---|
| $f = 250(03.1\%)$ | -0.61 | -0.65 | -0.57 | -0.86 | -1.23 | -6.33 |
| $f = 500(06.3\%)$ | -0.71 | -1.22 | -0.64 | -0.51 | -2.69 | -0.64 |
| $f = 250(06.8\%)$ | -0.40 | -1.16 | -0.26 | +0.11 | -1.08 | -0.70 |
| $f = 500(13.6\%)$ | -2.15 | -2.33 | -1.01 | +0.38 | -3.15 | -1.40 |

