# OpenReview forum: "Adversarial Attacks on Node Embeddings"
_ICLR.cc/2019/Conference_

### Official Review · AnonReviewer2 · 2018-11-02
**a novel adversarial attack on node embedding method based on random walks**

**Rating:** 6
**Confidence:** 4

**Review:**

Brief Summary:
The authors present a novel adversarial attack on node embedding method based on random walks. They focus on perturbing the structure of the network. Because the bi-level optimization problem can be highly challenging, they refer to factorize a random walk matrix which is proved equivalent to DeepWalk. The experimental results show that their approach can effectively mislead the node embedding from multi-classification.

Quality:
This paper is well-written except for some minor spelling mistakes

Clarity:
This paper is quite clear and easy to follow.

Originality:
This work follow the proposal Qiu et al.(WSDM'18)'s proof and present a novel approach to calculate the loss when the network changes by A'.

Pros:
1. Detailed proofs presented in appendix
2. They present 6 questions and answer them with effective experiments.
3. They present a new way to attack node embedding by factorizing a equivalent matrix.

Cons:
1. I have noticed that Zügner et al.(KDD'18) present an adversarial attack method on GCN for graph data. I think it is reachable by the day the authors submitted this paper. This is opposite to the first sentence "Since this is the first work considering adversarial attacks on node embeddings there are no known
baselines" said in Section 4.
2. The author present the time analysis of their approach but the efficiency result of their approach is not presented.
3. To enforce misclassification of the target node t, the author set the candidate flip edges as edges around t. Does this mean only the node's local edges can mislead the target node from downstream tasks? I think the authors should consider more candidate edges but this may lead to larger time complexity.
4. Figure. 4 tells that low-degree nodes are easier to mis-classify. If the baseline method B_{rnd} randomly select edges to flip among the local area of node t, I think the result should be similar to the proposed approach on low-degree nodes because the flipped edges should be the same.

== I have read the rebuttal. Thanks for the response.

---

> ### Author Response · Authors · 2018-11-19
> **Re: AnonReviewer2**
>
> Thank you for your review and feedback.
>
> 1) We discuss the work of Zügner et al. (KDD'18) in the fourth paragraph of section 2. While they are the first work to consider attacks on graphs they focus on targeted attacks for semi-supervised graph classification. In contrast, our work focuses on unsupervised node embeddings for which indeed there were no known attacks.
>
> 2) For the examined datasets the wall-clock time for our approach is negligible: on the order of few seconds when calculating the change in eigenvalues, and on the order of few minutes when calculating the change in eigen-vectors. Furthermore, the approach is trivially parallelizable which we have not yet exploited.
>
> 3) We restricted our experiments to candidate flips (i.e. edges and non-edges) around a target node t since initial experiments showed that they can do more damage compared to candidate flips in other parts of the graph. This intuitively makes sense since the further away we are from node t we exert less influence on it. Zügner et al. (KDD'18) show similar results (e.g. see their indirect attack). Note that for the general (non-targeted) attack all edges/non-edges are viable candidates.
>
> 4) We already make sure that B_{rnd} randomly selects edges/non-edges to flip among the local area of node t. Specifically, the candidate set from which we randomly sample is {(v,t) | v!=t} same as with our approach. Note that since we flip (d_t+3) edges/non-edges per target node t there are “(n-1) choose (d_t+3)” potential random outcomes for B_{rnd} which is relatively large even for small degree nodes. This implies that it’s unlikely to randomly select the same pairs as our method.

---

### Official Review · AnonReviewer1 · 2018-11-05
**An interesting but somewhat incomplete paper on poisoning graph embeddings.**

**Rating:** 6
**Confidence:** 3

**Review:**

This paper is a timely work on poisoning random walk based graph embedding methods. In particular, it shows how to derive a surrogate loss function for DeepWalk. Even though the analysis method and the algorithm proposed is somewhat loose, I think this paper do a good contribution towards the adversarial attack problem for important models.

There are still some room to improve in this paper. One problem is that since the paper proposes a surrogate loss L_{DW3}, it would be natural to analysis its gap from L_{DW1}. In this paper I can only see some empirical results on this issue. Another issue is that the algorithm the paper used is another approximation towards L_{DW3} by an additional sampling method. And the overall strategy can be far away from the true optimal solution for maximizing L_{DW3}. Still there's no analysis on that issue. A potential drawback for the method proposed is that its complexity is O(NE), which can be quite expensive when the graph is big, and # edges is \Omega(NlogN).

The experiments in this paper is convincing. It seems that the method proposed is way better than its competitors when removing edges. Is there any further intuition on that? Why the method is not so good when we add edges? Moreover, the black-box attack scenario requires more justification. What is the relative performance gain for A_{DW3} against other attacks in the black-box setting?

Overall, this paper targets on an important issue in machine learning. My main concern is that it leaves too many questions behind their algorithms. Still some effort is required to improve the paper.

---

> ### Author Response · Authors · 2018-11-19
> **Re: AnonReviewer1**
>
> Thank you for your review and feedback.
>
> L_{DW3} is an upper-bound on L_{DW1} so maximizing L_{DW3} is principled. We currently do not have any theoretical results on the tightness of the bound, however we added further experimental evidence in the appendix (Section 6.3 in the revised paper).
>
> In the revised version we now discuss the approximation via sampling you mentioned (Section 3.3 - The overall algorithm). Note that the sampling is only done for adding, while edge removal is always done without sampling. One reason why the method performs better when removing edges compared to adding edges could be exactly this further approximation. To reduce this effect one can easily invest more computational resources and simply increase the size of the candidate set. The approach is also trivially parallelizable.
>
> We added an analysis of the performance gain for A_{DW3} against other attacks in the black-box setting in the appendix (Section 6.3 in the revised paper). A_{DW3} is able to cause more damage compared to the baseline attacks.

---

### Official Review · AnonReviewer3 · 2018-11-09
**Nice first try but needs improvement**

**Rating:** 5
**Confidence:** 5

**Review:**

The topic of this paper is interesting; however, the significance of the work can be improved.  I recommend that the authors test the vulnerability of node embeddings on various random graph models.  Examples of random graph models include Erdos-Renyi, Stochastic Kronecker Graph, Configuration Model with power-law degree distribution, Barabasi-Albert, Watts-Strogatz, Hyperbolic Graphs, Block Two-level Erdos-Renyi, etc.  That way we can learn what types of networks are more susceptible to attacks on random-walk based node embeddings and perhaps look into why some are more vulnerable than others.

---

> ### Author Response · Authors · 2018-11-19
> **Re: AnonReviewer3**
>
> The suggestion to test the vulnerability of node embeddings on various random graph models and examine if some are more vulnerable than others is potentially interesting. However, this is a completely different objective than the one discussed in our paper. Furthermore, our experiments on real-world graphs yield arguably more relevant insights than experiments on synthetic graphs.

---

### Meta-Review · Area_Chair1 · 2018-12-19
**Novel analysis of an important problem, but needs some improvements**

**Confidence:** 3
**Recommendation:** Reject

**Metareview:**

The paper provides a novel analysis of the robustness to adversarial attacks in network representation learning. It appears to be a useful contribution for important class of models; however,  the detailed reviews (1 and 2) raise some concerns that may require a bit of further work (though partially addressed in revised version).